# A Deterministic Branch Prediction Technique for a Real-Time Embedded Processor Based on PicoBlaze Architecture

**Ehsan Ali and Wanchalerm Pora ***

Department of Electrical Engineering, Faculty of Engineering, Chulalongkorn University,
Bangkok 10330, Thailand
*   Correspondence: wanchalerm.p@chula.ac.th

**Abstract:** This paper proposes a new deterministic branch prediction unit to achieve a uniformly timed instruction set architecture (ISA). The deterministic ISA is achieved by utilizing two address buses in conjunction with dual-port block RAMs that are common in commercial FPGAs. The goal is to remove mandatory branch and load delays to achieve a uniform one clock cycle per every instruction. To demonstrate the concept, the proposed architecture is applied to the Xilinx PicoBlaze firm core. The result is a new soft core named DAP-Zipi8 that reduces the clock per instruction (CPI) metric of PicoBlaze from two to one at the expense of extra logic and a longer critical path. The increased critical path reduces maximum achievable clock speed from 357.509 MHz to 224.022 MHz. Merging the gain in CPI with the loss in maximum clock frequency still improves overall processor performance by 18.28–19.49%. The high-performance deterministic DAP-Zipi8 is a viable choice for hard RTES applications.

**Keywords:** FPGA; field programmable gate arrays; microprocessors; real-time embedded systems; Xilinx PicoBlaze; deterministic instruction set architecture

## 1. Introduction

This paper focuses on central processing units for real-time embedded systems (RTESs). The majority of microprocessors available on the market are not designed for hard RTESs [1]. Advanced performance improvement techniques (pipelining, branch prediction units (BPUs), floating point units (FPUs), caching, memory management units (MMUs), frequency scaling, shared buses, etc.) sacrifice determinism and introduce *timing anomalies* [1–3] which increase the complexity of static timing analysis (STA) [4,5].

A good example of the increase in the complexity of STA is the case of a *pipeline stall*, where execution of an instruction must stall (e.g., due to register data dependency) for $n$ extra cycles where $n$ depends on pipeline depth. Another example is incorrect predictions from the BPU, which forces the processor to discard speculatively fetched instructions, thus incurring a delay (equal to the number of stages between the fetch and execute stages [6]).

FPU performance depends on implementation and input operands. For example, a subnormal input can increase the execution time by two orders of magnitude [7]. A cache miss requires the upper memory layers to be accessed, which imposes a much longer delay. Accessing a memory page that is not mapped into virtual address space causes a page fault in the MMU, forcing a page to be loaded from disk which, again, incurs a delay. Frequency scaling and shared buses exhibit similar non-deterministic delays. All these performance improving techniques introduce timing anomalies and increase STA's complexity.

There is a misconception that fast computing equals real-time computing. Rather than being fast, the most important property of RTESs is predictability [8]. All tech-

niques mentioned above are sources of indeterminism. They add complexity to static analysis tools and have a negative impact on worst-case execution time analysis (WCET), which determines the bounded response time of an RTES. Although achieving acceptable WCET analysis is still possible in the presence of those advanced techniques (through end-to-end testing, static analysis, and measurement-based analysis [9]), achieving better WCET analysis when some features are present (e.g., caches [1]) is still an open problem. Therefore, designers tend to use simpler microprocessors that have adapted reduced instruction set computer (RISC) architecture with less of those performance improving features for hard real-time systems. The RISC architecture has a major advantage in real-time systems as the average instruction execution time is shorter than complex instruction set computer (CISC) architecture. This leads to shorter interrupt latency and shorter response times [10]. One of the major neglected sources of performance inconsistency is indeterministic instruction set architecture (ISA). Branch instructions require more clock cycles if taken than not taken. For example, ARM11 branch instructions require three clock cycles if taken, but one cycle if not taken [11]. In PowerPC 755, a simple addition may take anywhere from 3 up to 321 cycles [12] due to its non-compositional architecture [13] that produces a domino effect.

For most 4-bit, 8-bit, 16-bit, and non-pipelined microarchitectures without caches, one could simply sum up the execution times of individual instructions to obtain the exact execution cycle of the instructions sequence [14,15]. This is only valid if the ISA of a microarchitecture is deterministic. In this context, determinism means the exact number of clock cycles for all instructions is known, and the number of clock cycles per instruction is permanent and does not vary based on previous states of the processor. This property is very important in hard real-time embedded systems that need to respond to external events (e.g., execution completion of machine instructions in a procedure) with precise timing. In those systems, WCET estimation cannot be used, as even a single clock cycle deviation from expected timing makes the system non-functional. A good example of such systems is the controller of multi-core architectures, where a complex finite state machine performs the role of an operating system and delegates independent tasks to cores and retrieves the result.

Consequently, RISC-V, ARM, Intel, MIPS, and all processors that have a pipeline, cache systems, or other sources of indeterminism cannot be used in systems where cycle-accurate predication is one of their hard requirements. PicoBlaze is a good choice as it is already a deterministic core (uniform CPI = 2) with relatively low performance. It can be used as a controller for a complex finite state machine that governs multiple cores.

In this paper, a technique for a deterministic branch prediction is proposed. Using the proposed design, the processor always has the correct program counter regardless of whether the branch is taken or not, which eliminates ISA indeterminism. The Xilinx PicoBlaze firm core has a clock per instruction (CPI) value of two for all its instructions [16]. It is already a deterministic core through the setting of CPI to two. It is modified to incorporate the proposed architecture in this paper. A lookahead circuit, in conjunction with a dual-fetch mechanism, is employed for reducing the CPI from two to one while retaining the ISA determinism (identical CPI for all instructions).

The uniform CPI = 1 value for all instructions is achieved by removing register data dependency and flags/conditional branch interlocks. That is why "branch and load delay" definitions are given; how other architectures have dealt with them will also be discussed. Note that CPI provides a sufficient way of comparing two different implementations of the same ISA (in our case PicoBlaze ISA) [17]; therefore, no benchmarking program is required because both cores execute the same instruction sequence.

The objective and contribution of our work is to improve processor performance without sacrificing ISA determinism. In the case of Xilinx PicoBlaze, the objective can be translated to improving the performance of the core from CPI = 2 to CPI = 1. A dual-fetch technique alongside a branch prediction circuit is proposed that fetches two instructions at one clock cycle and uses the second fetch for the sole purpose of removing branch and

load delays with the goal of achieving uniform CPI = 1 values. The dual-issue technique (related work) requires a pipeline and refers to fetching two instructions at each clock cycle and then issuing them to the next stage of a pipeline to achieve CPI = 0.5 without a guarantee of CPI uniformity. In our ongoing project, a complex finite state machine has been implemented using a PicoBlaze core that controls 1024 other PicoBlaze cores. Because of deterministic ISA, the state machine can react to external triggers, such as completion of procedure execution, and can retrieve and then pass the result to other cores at precise clock cycles (precise timing).

The contributions of this paper are:

1. A microprocessor architecture that eliminates branch and load delays to achieve uniform CPI = 1 values.
2. The utilization of unused ports of FPGA memory primitives to boost overall processor performance while retaining ISA determinism.
3. The 18.28–19.49% performance improvement of Xilinx PicoBlaze in terms of MIPS.

Preliminary definitions are provided in the next session and related work is presented in Section 3. A brief overview of PicoBlaze architecture is then provided in Section 4. In Section 5, a technique (proposed in [18]) is employed to transform the PicoBlaze into a modifiable soft core named Zipi8. The source code of the new core is written at the RTL-level, which makes architectural customization possible. Section 6 discusses the Zipi8 modifications used to achieve CPI = 1; the modified core is named DAP-Zipi8. The work presented in this section contains the two main contributions of the paper. Finally, the comparison of resource and power utilization for DAP-Zipi8 versus PicoBlaze is presented in Section 8. The verification process is covered in Section 9.

## 2. Definitions

Real-time systems (RTSs) are computing systems that must react within precise time constraints to events in the environment [19]. We can categorize RTSs into three groups [18]:

1. **Hard RTSs**: impose strict timing requirements with fatal consequences if temporal demands are not met.
2. **Soft RTSs**: set coarse temporal requirements, without catastrophic consequences if several deadlines are missed.
3. **Firm RTSs**: set fine-grained temporal requirements, without fatal consequences in the case of infrequent deadline misses.

Embedded systems are computing systems with tightly coupled hardware and software integration that are designed to perform a dedicated function [20]. The reactive nature of embedded systems is shown in Figure 1. A reactive system must respond to events in the environment within defined time constraints. External events being aperiodic and unpredictable makes it more difficult to respond within a bounded time frame [21].

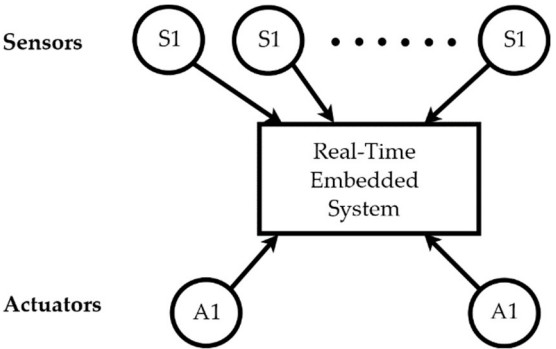

**Figure 1.** A model of sensors and actuators in an embedded system [21].

Hard real-time embedded systems (**RTESs**) refer to those embedded systems which require real-time behavior with for a missed deadline [22]. The software part of an RTS is an application that runs either in stand-alone mode (bare metal) or scheduled as a task on a real-time operating system (RTOS). The hardware part includes one or more central processing units (CPU), memory elements, and input/output (I/O) devices with interrupt mechanisms to provide deterministic bounded responses to external events.

The term **timing anomaly** refers to a situation where a local worst case does not entail the global worst case. For instance, a cache miss (the local worst case) may result in a shorter execution time than a cache hit due to scheduling effects [3]. The **domino effect** is a severe special case of timing anomalies that causes the difference in execution time of the same program starting in two different hardware states to become arbitrarily high [13].

One of the metrics of microprocessor performance is the average number of clock cycles per instruction (**CPI**), the lower the value the better the performance. Given a sample program with $p$ instructions, the instruction count $n_i$ for each instruction type $i$, and the number of clocks needed to execute instruction type $c_i$, CPI can be defined as shown in Equation (1).

$$CPI = \frac{\sum_i n_i c_i}{p} \tag{1}$$

CPI in conjunction with processor clock rate can be used to determine the time needed to execute a program [17]. The classic 8051 CPU requires at least 12 cycles per instruction (CPI > 12) [23], PIC16 takes 4 cycles or more (CPI > 4) [24], but Xilinx PicoBlaze takes 2 clock cycles exactly (CPI = 2) [16]. Optimization of CPU architecture may achieve CPI = 1 for most instructions, but a few of them still need more than one cycle. This takes away the ISA *uniformity* attribute.

The implementation of processor-based design can be done via three mediums:

1. A Microcontroller Unit (MCU).
2. A Field-Programmable Gate Array (FPGA).
3. An Application-Specific Integrated Circuit (ASIC).

We exclude the Application-Specific Integrated Circuit (ASIC) as an approach due to its high Non-Recurring Engineering (NRE) cost, and its impracticality for low volume production [25].

An FPGA chip includes input/output (I/O) blocks and a core programmable fabric [26]. FPGAs are being used extensively to cover a broad range of digital applications, from simple glue logic [27] and hardware accelerators to very powerful System-on-Chip (SoC) platforms [28]. Having an 8-bit architecture as the cornerstone of MCUs used in designing tiny embedded systems is widely accepted [29].

FPGAs have a higher level of flexibility than MCUs by providing a programmable logic (PL) fabric [30]. For example, FPGAs allow designers to change a product after release by upgrading its firmware [31]. The drawback of FPGA flexibility is that it uses approximately 20 to 35 times more area, has a speed roughly 3 to 4 times slower, and consumes roughly 10 times as much dynamic power [32]. There are also occasions when FPGAs can outperform MCUs by implementing kernel applications in PL and integrating them with soft cores [33] to take advantage of the inherent parallelism of FPGA devices in an optimal way [34]. Meanwhile, FPGAs can host intellectual property (IP) CPU cores with the capacity to add custom instructions (e.g., Nios-II [35]).

IP cores come in three different flavors [36]:

1. **Soft cores**: written in HDL without extensive optimization for the FPGA target architecture.
2. **Firm cores**: written in HDL but implementations have been optimized for a target FPGA architecture.
3. **Hard cores**: fixed-function gate-level IP within the FPGA fabric.

One of the important applications of IP cores is in safety-critical real-time embedded systems where designers can take advantage of deterministic timing [37–40]. The Xilinx PicoBlaze is a firm core with uniform CPI = 2 values, which results in deterministic ISA performance [16]. Additionally, it is an industry-level core with enough users to find and fix its potential bugs. Unfortunately, its behavioral HDL source code is not available. The available source code is highly optimized and uses Xilinx primitives. The optimized design has a very small footprint on the FPGA fabric, but its modification and implementation on non-Xilinx devices is nearly impossible.

*Performance versus Determinism*

Three factors contribute to system performance:

1. The no. of instructions required to perform a task ($I$).
2. The no. of clock cycles required per instruction ($CPI$).
3. The period of a clock cycle ($T$).

Both RISC and CISC attempt to minimize $T$. For CISC, the emphasis is on minimizing $I$ by providing powerful instructions. This results in an increase in $CPI$. For RISC, the goal is to minimize $CPI$, and to bring the $CPI$ value as close as possible to one [41,42].

To achieve CPI = 1, RISC processors resort to the pipelining technique. The major problem with pipelined architectures is that if arbitrary instruction $B$ in the pipeline has data dependency with its previous instruction $A$, then the pipeline must be stalled until instruction $A$ passes the execution stage. This delay is called *load delay*. Most RISC processors are designed to possess a load delay of one clock cycle (introducing a load delay slot [41]) but come short of eliminating it entirely.

Another similar case applies to conditional branch instructions. They depend on flags set by previous instructions. Therefore, the pipeline must be stalled to let previous instructions finish, and then the decision to whether the branch must be taken or not can be made. This hold up time is called a *branch delay*.

Another issue is when a taken branch invalidates the next immediate fetched instruction in the pipeline and forces a flush. There are two solutions to this:

1. Insert a No Operation (NOP) instruction after each branch instruction.
2. Always execute the instruction after the branch even if the branch is taken (delayed branch [41]).

This instruction slot that gets executed without the effect of previous instructions is called the *branch delay slot*. To help clarify these concepts they are concisely depicted in Figure 2.

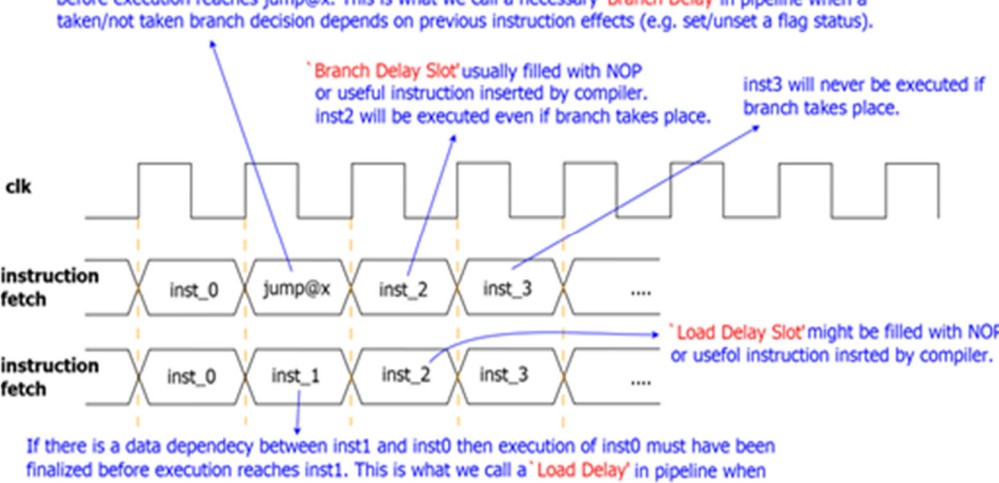

**Figure 2.** Branch and load delay—branch and load delay slots.

Aside from advances in fabrication, the common way to speed up the clock is to chop a pipeline into many stages (deep pipeline) [43]. Modern processors departed from classic 5-stage pipelines and went up to 50 stages [44,45]. However, when power was considered [46], the dynamic and leakage power per latch suggested an optimal pipeline depth of around 14 to 20 stages [47]. As the number of pipeline stages increases, the stalls become more costly. To minimize stalls, several techniques, such as branch prediction, were introduced which worsen the determinism of microarchitectures. For example, the core Intel i7 pipeline with 14 stages imposes a cost of an extra 17 clock cycles when a branch misprediction occurs [47].

The search for RTES microprocessors yields no definite results, as all modern processors have deviated from simple architectures and have added performance improving features. In practice, designers choose a very high performance *indeterministic* processor to meet the WCET requirement. Even processors such as those of the ARM Cortex-R series(advertised as real-time processors) carry the inherent indeterminism concepts discussed earlier. For example, Cortex-R4 branch instruction may take one, eight, or nine clock cycles based on correct/incorrect dynamic prediction results [48]. What differentiates the Cortex-R series (R stands for real-time) processors from general-purpose processors is tightly coupled memory with error correction code (ECC) employed, redundant lock-step configuration with logic for fault detection, and a low latency deterministic interrupt system that allows multi-cycle instructions to be interruptible and avoid cache misses in memory management units.

Considering the arguments presented above, we suggest that, since predictability is more important than performance in RTESs, there are situations where a hard RTES prefers a low-power, non-pipelined, non-cache microprocessor that enjoys a deterministic ISA over a high-performance processor with an indeterministic ISA, even if that processor has a pipeline and cache. In those cases, our proposed DAP-Zipi8 processor, which is Xilinx PicoBlaze compatible, can be utilized.

## 3. Related Work

Simple architectures, such as the binary decision machine (BDM) [49], can achieve CPI = 1 because they do not have branch instructions [50]. BDMs for complex tasks that support a limited number of instructions working on the data path, plus 'call' and 'return' instructions to support subprograms, are also proposed. Although they achieve a RISC-like behavior with CPI = 1, they still lack conditional branches [51].

Non-pipelined single-cycle processors are widely used in academia for teaching processor architecture (such as MIPS and single-cycle RISC-V [17,52]). Although their CPI is one, their clock period is very long, which makes them inefficient [17]. This forces techniques such as pipelining to be used to shorten the clock period. A pipelined processor can only achieve CPI = 1 (an idealized goal) if all instructions are independent [53].

Table 1 lists several pipelined RISC processors and the solution that each has adopted to deal with load and branch delays. The picks are based on historical importance: the IBM 801 resulted in PowerPC [54], Berkeley RISC-1 contributed to SPARC [55], and Stanford RISC developed into MIPS [56]. ARM and RISC-V are also recent notable architectures. All these processors have non-uniform instruction timing which contributes to indeterminism.

**Table 1.** RISC Solutions to Load and Branch Delays.

| Processor | Load Delay | Branch Delay |
|---|---|---|
| IBM 801 | Locks register, can be optimized by a compiler [57] | Branch with execute (BWE) [57] * |
| RISC I | Load and Store, always takes two cycles [42] | Delayed jump [42] † |
| SPARC-V8 | Load-use interlock stalls the pipeline [58] | Annulling delayed branches [58] ‡ |
| SPARC-V9 | Similar to SPARC-V8 but a 64-bit version | Annulled delayed branches [59] ‡ |
| MIPS-I | Delayed loads with mandatory load delay slot [60] | Delayed branch with a branch delay slot [60] |

| MIPS-II | Removes mandatory load delay slot; in case of violation, extra real cycles will be added [61] | Branch-likely [62] § |
|---|---|---|
| MIPS32 | Interlock by load delay stalls the pipeline [61] | Branch-likely, compact branches [63] ‖ |
| ARM7TDMI (3-stages) | All loads take at least three cycles [64,65] | All branches take at least three cycles [64,65] |
| ARM9TDMI (5-stages) | Load-use interlock incurs one extra cycle if the following instruction uses a loaded word [66] | All cases take three cycles [64,66] |
| ARM11 (8-stages) | Takes one to five clock cycles due to register interlocks [11] | Dynamic branch prediction/folding [11] ¶ |
| SiFive E31 (RISC-V) | All loads have three-cycle result latency [67] | Branch predictor with one-cycle latency, misprediction incurs an extra three cycles [67] |
| PowerPC 750 CL | Out-of-order load/store unit with two or three cycles of latency | Static/dynamic branch prediction/folding ∗∗ |

* Executes the instruction in the branch delay slot even if the branch is taken. In total, 60% of instructions can be converted to execute form by the compiler. † Delayed jumps are for every branch with compiler optimization to either insert a NOP after each branch or a safe instruction. ‡ If the branch is taken, it always executes the instruction in the delay slot; if not taken, then it checks the annul bit. If the annul bit is 1, annul the instruction in the slot. If it is 0, then execute it. Using the annul bit, compiled code contains less than 5% NOP. § Branch-likely is similar to annulling/annulled delayed branches. ‖ Prior to release six: has a branch delay and uses branch-likely instructions. Release six: no delay slot and uses compact branches which have a forbidden slot instead. Adjacent control transfer instructions (CTIs) introduce a performance penalty. An untaken branch requires one cycle, and a taken branch requires three or more cycles. ∗∗ Branch instruction gets folded if taken (needs no cycle), and one idle cycle will be added on branch target instruction cache (BTIC) miss. The pipeline gets flushed on branch misprediction (takes three cycles or more).

The effect is amplified when performance improving techniques such as caches, dynamic branch prediction, or branch folding are present. For example, in PowerPC 750 CL, the timing for branch instruction is highly irregular and is based on [67]:

- Whether the branch is taken;
- Whether instructions in the target stream are in the branch target instruction cache (BTIC);
- Whether the target instruction stream is in the cache;
- Whether the branch is predicted;
- Whether the prediction is correct.

This shows an extreme level of indeterminism which ultimately makes calculation of WCET more complex. There are also unconventional ways of achieving a CPI of one, such as CoolRISC [68,69] which uses a double-latch clocking scheme with two non-overlapping clocks to eliminate load and branch delays. The drawbacks of this approach are:

1. Incompatibility with optimization algorithms embedded in electronic design automation (EDA) tools.
2. No FPGA primitive support to implement the design.
3. Accessing memory after MUL instruction needs two cycles instead of one, and interrupt and events have a delay in some cases.
4. Difficulty reaching high clock speeds (e.g., 60 MIPS needs a 120 MHz oscillator).

## 4. The PicoBlaze Firm Core

### 4.1. Overview

KCPSM6 is an upgraded version of the (K)constant Coded Programmable State Machine 3 (KCPSM3) [70] and is the technical name of Xilinx PicoBlaze. It is an 8-bit firm core with 32 general-purpose 8-bit registers which are arranged in two banks. All instructions have 18-bit width and need two clock cycles to be executed [16]. The instruc-

tion bitfields are divided into a 6-bit opcode allowing $2^6 = 64$ unique instructions (55 out of 64 instructions are implemented) and a 12-bit field set aside for operands, as shown in Table 2. The core architectural overview is shown in Figure 3. Its program memory can go up to 4 KB and it has scratch pad memory (SPM) for temporary data storage, with a maximum size of 256 bytes. Additionally, it has a stack with a depth of 30 and 256 I/O ports.

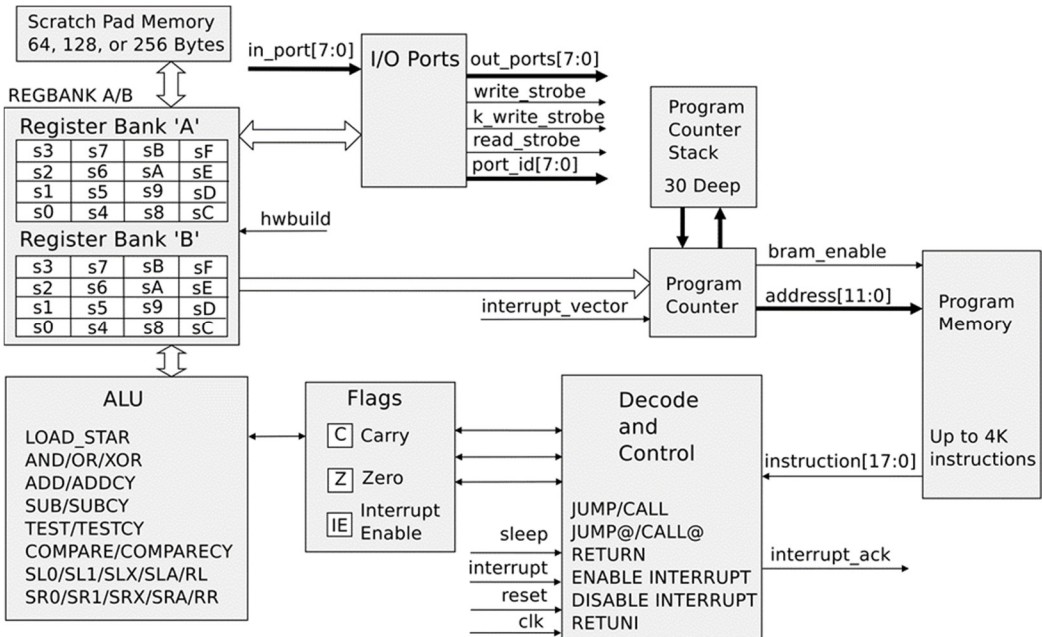

**Figure 3.** KCPSM6 architecture and features [16].

As shown in Table 2, the 12-bit operand field accommodates one or a mixture of the following values: "*aaa, kk, pp, p, ss, x, y*". For example, the "JUMP *aaa*" instruction is encoded to a 0x22*aaa* hex value, 0x22 is the opcode, and 0x*aaa* is the 12-bit jump target address, or "LOAD sX, sY" is encoded to 0x00*xy*, 0x00 is the opcode, 4-bit *x* is the destination register, and 4-bit *y* is source register. PicoBlaze has three flags: Carry (C), Zero (Z), and Interrupt Enable (IE). There is an interrupt pin which forces the processor to execute code residing in the Interrupt Service Routine (ISR) (its memory address location is predefined), and there is a sleep input pin for freezing all operations [16].

**Table 2.** PicoBlaze Instruction Bitfields [16].

| Opcode (6-bit) | | Operands (12-bit) |
| --- | --- | --- |
| 6-bit always | *aaa* | 12-bit address (0x000-0xFFF) |
| | *kk* | 8-bit constant (0x00-0xFF) |
| | *pp* | 8-bit port ID (0x00-0xFF) |
| | *p* | 4-bit port ID (0-F) |
| | *ss* | 8-bit scratch pad location (0x00-0xFF) |
| | *x* | 4-bit register within the bank (s0-sF) |
| | *y* | 4-bit register within the bank (s0-sF) |

### 4.2. PicoBlaze Source Code Analysis

The PicoBlaze core is provided in both VHDL and Verilog languages. VHDL is chosen for describing the proposed hardware design. FPGA primitives are the basic building blocks of a design. They perform dedicated functions in the device, implement standards for I/O pins, and have standardized names [71].

The first step in source code analysis is to scan the code for all primitives used in the design. The list of all primitives used in PicoBlaze is as follows: "LUT6, LUT6_2, FD, FDR, FDRE, XORCY, MUXCY, RAM32M, RAM256X1S".

The second step is to study the FPGA manufacturer's library guide to retrieve the detailed functionality of each primitive, and then write a VHDL implementation of it to obtain vendor-independent modules [72]. In the case of PicoBlaze, the "Xilinx 7 Series FPGA Libraries Guide" [73] provides the detailed behavior of each primitive. The next section provides the equivalent *vendor-independent* VHDL code of each primitive.

## 5. Zipi8: A PicoBlaze Compatible Soft Core

In this section, the methodology behind transforming a PicoBlaze firm core to a soft core using vendor-independent primitive definitions (in VHDL) is detailed.

### 5.1. Primitive Conversion to Vendor-Independent VHDL

One of the primitives listed in the previous section is picked as an example: LUT6. The Xilinx Library Guide reads "LUT6 is a six-input look-up table (LUT), it can either act as asynchronous 64-bit ROM (with 6-bit addressing) or implement any six-input logic function" [73]. A VHDL implementation must be written according to the extracted definition of the primitive.

Listing 1 shows one of the LUT6 instances used in the PicoBlaze core as an example. The 'pc_mode2_lut' is the instance name, and 0xFFFF_FFFF_0004_0000 is a 64-bit hexadecimal constant used as the initial value of the LUT6 primitive. I0, I1, I2, I3, I4, and I5 are inputs, and O is output signals.

First, a Boolean function minimization on the six-input logic function using the given 64-bit LUT value is performed. The minimization method can be either manual or automated, using algorithms such as the Espresso logic minimizer [74]. Equation (2) shows the result of minimization of the six-input logic function LUT6(I5, I4, I3, I2, I1, I0) shown in Listing 1.

$$LUT6(I5, I4, I3, I2, I1, I0) = O = I5 + I4 \cdot \overline{I3} \cdot \overline{I2} \cdot I1 \cdot \overline{I0} \tag{2}$$

**Listing 1:** An example of LUT6 primitive instantiation used in the PicoBlaze core.

```
pc_mode2_lut : LUT6
generic map (INIT=>X"FFFFFFFF00040000")
port map (
      I0 => instruction (12),
      I1 => instruction (14),
      I2 => instruction (15),
      I3 => instruction (16),
      I4 => instruction (17),
      I5 => active_interrupt,
      O => pc_mode (2)
);
```

After replacing the I0, I1, I2, I3, I4, I5, and O variables in Equation (2) with the name of signals connected to them, the exact equivalent vendor-independent VHDL implementation of LUT6 can be derived, as shown in Listing 2.

**Listing 2:** An example of vendor-independent VHDL implementation of LUT6.

```
pc_mode (2) <=
        active_interrupt or
        instruction (17) and
        (not instruction (16)) and
        (not instruction (15)) and
    instruction (14) and
        (not instruction (12));
```

The case for other primitives is the same. The vendor-independent VHDL implementation of the rest of the primitives, including "LUT6_2, FD, FDR, FDRE, XORCY, MUXCY, RAM32M, RAM256X1S", can be found in Supplementary S1, which includes the VHDL source code of all primitives in a Xilinx Vivado project.

*5.2. Modular Conversion of PicoBlaze to Zipi8*

The PicoBlaze VHDL source code has no modular structure. It is a module in a VHDL file with a long list of primitive instantiations connected via signals. To convert the design from a firm core (PicoBlaze) to soft core (named Zipi8 by the authors), it is sufficient to directly replace all the instances with vendor-independent VHDL equivalent code, as mentioned in the previous section. If, along the process, the related primitives are grouped into VHDL modules (based on the characteristic equation of flip-flops) and then transformation is performed, then complexity can be managed, human errors are minimized, and a modular design emerges. Additionally, the process provides better understanding of the internal architecture of the design.

The PicoBlaze core is transformed into 16 modules which use source code comments and original primitive names. The module names are listed below, and their source code can be found in Supplementary S1:

1.  arith_and_logic_operations;
2.  decode4alu;
3.  decode4_pc_statck;
4.  decode4_strobes_enables;
5.  flags;
6.  mux_outputs_from_alu_spm_input_ports;
7.  program_counter;
8.  register_bank_control;
9.  sel_of_2nd_op_to_alu_and_port_id;
10. sel_of_out_port_value;
11. shift_and_rotate_operations;
12. spm_with_output_reg;
13. stack;
14. state_machine;
15. two_banks_of_16_gp_reg;
16. x12_bit_program_address_generator.

The modules listed above and important signals between them are shown in Figure 4. It is a simplified version of a fully detailed schematic that is available (in Supplementary S2) in Encapsulated Postscript (EPS) format. To simplify the diagram, occasionally two or three related modules are combined. This is indicated by mentioning module numbers in parentheses. For example, the 'Decoders' module consists of three submodules: (2), (3), and (4). Both program memory and the processor share the same clock signal. Those modules which are synchronized with the clock are marked with a triangular

symbol. The absence of a clock symbol indicates pure Combinatorial Logic (CL) (e.g., the 'Operand Selection' module).

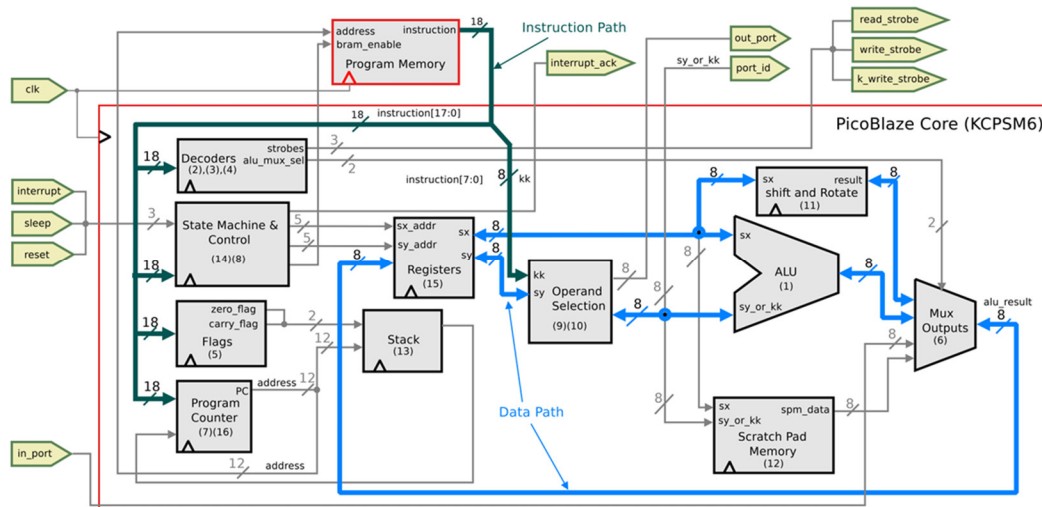

**Figure 4.** Block diagram of modular Zipi8 (a high-resolution version of this figure is available in Supplementary S3).

### 5.3. Zipi8 Architecture

The important paths, such as the 'data path' and 'instruction path', are explicitly marked in Figure 4. The allocation of two separate buses connected to two different memory blocks indicates a Harvard architecture [56]. To explain the instruction execution mechanism of PicoBlaze, a sample program (Listing 3) with a branch instruction is manually traced.

**Listing 3:** A sample PicoBlaze program.

```
Start_at_0x000:
     LOAD  s0, 05          ;Loads value 05 into registers 0 – Mem. Location: 0x001
     LOAD  s1, 04          ;Loads value 04 into registers 1 – Mem. Location: 0x002
     JUMP   subprogram_at_01c              ; – Mem. Location: 0x003
     ; ...
     subprogram_at_01c:
     ADD   s1, s0          ; s1 <= s1 + s0              ; – Mem. Location: 0x01c
```

As shown in Figure 5, the de-assertion of the *reset* signal puts the processor into the *run* state. In this state, the processor waits for the first rising edge of the clock that triggers an instruction fetch from memory location 0x000. The fetch results in the 'Instruction Path' bus (see Figure 4) hold valid data (it is the first instruction, 'LOAD s0, 05', in Listing 3).

The instruction bus is connected to flip-flops in 'Decoders', 'State Machine & Control', 'Flags', and 'Program Counter' modules. When the second clock arrives, the instruction is decoded (sx_addr is set to 0 to select register s0, and the 05 constant value is placed on the instruction [7:0] bus, the *kk* instruction bitfield), the next state of machine is calculated, flags are set, and finally the program counter (PC) is incremented by one.

In the third clock cycle, the instruction at location 0x001 is fetched and the result of the ALU is written back into the register in parallel. This results in the *s0* register hold-

ing the constant value 05. In the next clock cycle, the instruction at location 0x001 (which is 'LOAD s1, 04') is fetched.

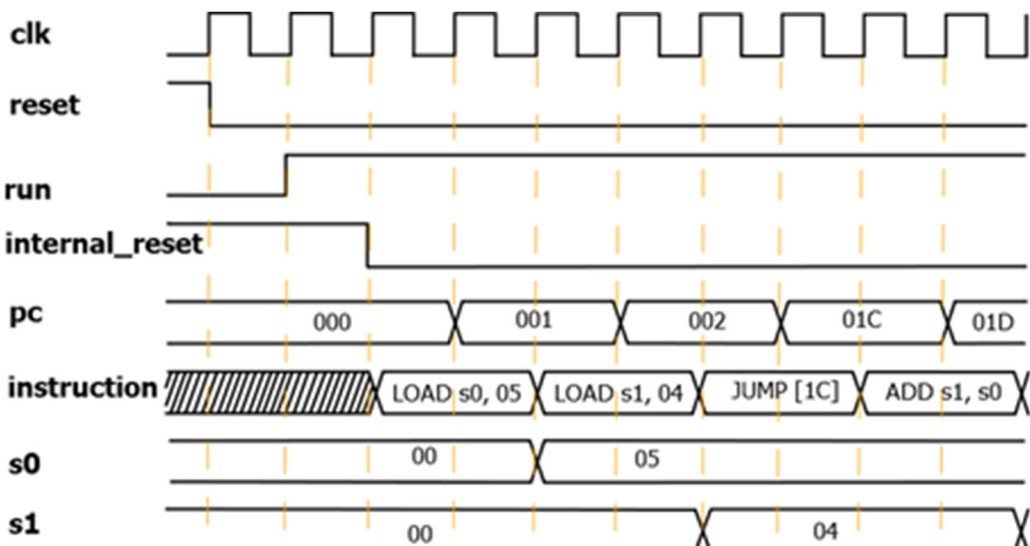

**Figure 5.** Tracing PicoBlaze instruction execution after a reset signal is asserted.

As with previous instructions, the decode and execute stages happen in the next clock cycle, which sets the *sx_addr* signal (see Figure 4) to 1 and prompts the second ALU operand (*kk* bitfield) to hold the constant value 04. In the next clock cycle, the processor writes back the result into the register bank, resulting in constant value 04 being stored in the *s1* register and, at the same time, the next instruction ('JUMP subprogram_at_01c') being fetched.

In the next cycle, the JUMP instruction is decoded and, instead of '*pc* = *pc* + 1' and the next consecutive instruction being fetched, *pc* is set to a value of 0x01C, which is the jump target location. In the next cycle, the instruction at location 0x01C of program memory ('ADD s1, s0') is fetched. The ADD instruction is then decoded, and the ALU needs some time (ALU propagation delay) to perform the add operation. The result is ready before the rising edge of the next clock cycle arrives, when it will be written back into the *s1* register, and so on. This manual execution tracing clearly shows the behavior of the PicoBlaze when it executes a branch instruction in two clock cycles.

Each original PicoBlaze instruction takes exactly two clock cycles (CPI = 2), making its ISA performance deterministic. This turns PicoBlaze into a suitable candidate for safety-critical real-time embedded systems [38] if its performance can be improved without adding a pipeline or caches. In the next section, a new design is proposed that achieves CPI = 1 with PicoBlaze, resulting in significant performance improvement.

### 5.4. Zipi8 Verification

We use the *comparison* method to verify the integrity of the Zipi8 core against the PicoBlaze. Flip-flop output signals have a one-to-one relationship in both cores. Therefore, the transformation process can be validated by probing signals at all output junctures of flip-flops in both cores and by using VHDL *assert* statements to catch any discrepancies between them. Verification details and extra information on PicoBlaze to Zipi8 conversion can be found in [18].

### 6. DAP-Zipi8: A Modified Zipi8 Soft Core with CPI = 1

Modification of the PicoBlaze becomes feasible after converting it to the Zipi8 soft core. The goal is to improve performance without those indeterministic performance improvement techniques that were discussed in the Introduction. In this section, at first, the

overall mechanism of the proposed technique, in terms of how to reduce CPI from two to one, is provided without diving deep into details. Next, as a case study, the proposed design is applied to the converted Zipi8 core, which is Xilinx PicoBlaze compatible.

*Branch and Load Delay Elimination*

Figure 6 shows how simultaneous fetching of two instructions per clock cycle eliminates branch delay. Assuming instructions placed in memory location 0, 1, 2, and 3 are named inst_0, inst_1, inst_2, and inst_3, then inst_0 and inst_1 are fetched simultaneously in the first clock cycle, inst_1 and inst_2 in the second cycle, and so on. If an instruction is a conditional jump to location $x$, then it is listed as jump@x.

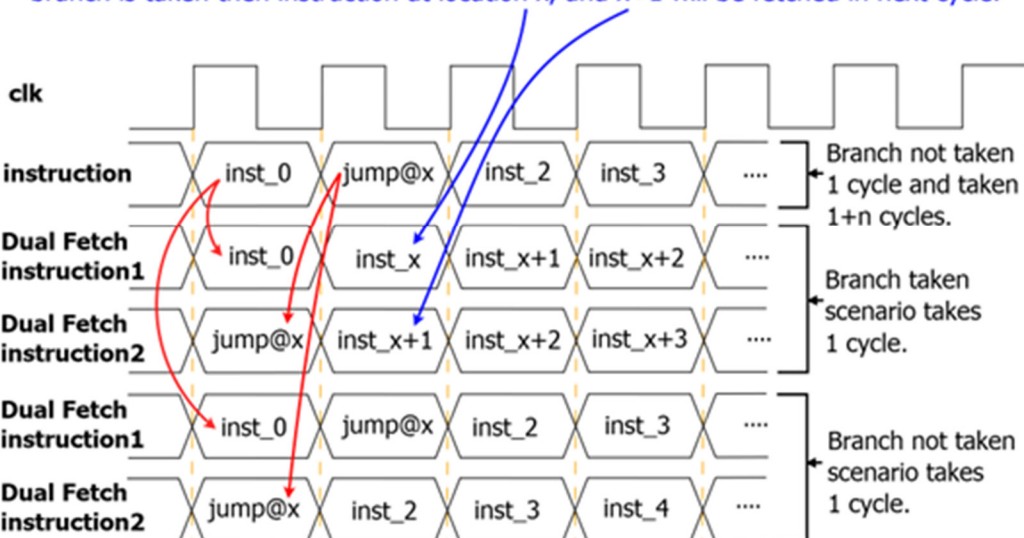

**Figure 6.** Description of the dual-fetch mechanism and how it allows conditional branch instructions to take one clock cycle, regardless of them being taken or not taken.

In Figure 6, inst_0 and jump@x are fetched in the first cycle (dual fetch); the second instruction is a conditional jump. Decoding both instructions simultaneously determines whether the conditional jump must be taken or not taken prior to the arrival of the second clock cycle. Knowing that a branch instruction will be taken or not taken is the cornerstone of the proposed design. In the second cycle, jump@x and inst_2 are fetched. If the branch is not taken (known in advance), then jump@x will be considered as a no operation (NOP) instruction and it takes one cycle to be executed. If the branch must be taken (known in advance), then, instead of fetching the second instruction (jump@x) in the second cycle, the instructions at location $x$ and $x + 1$ will be fetched simultaneously (inst_x and inst_x + 1) by spending only one clock cycle. Applying the proposed dual-fetch design to predict the behavior of branch instructions results in conditional instructions taking one cycle, whether they are taken or not taken.

The term *dual-fetch* should not be confused with the *dual-issue* feature that exists in some modern processors, such as the ARM Cortex-R. The dual-fetch technique proposed in this paper fetches two instructions in one clock cycle and uses the second fetch for the sole purpose of removing branch and load delays with the goal of achieving uniform CPI = 1 values. The dual-issue feature requires a pipeline and refers to fetching two instructions in each clock cycle, and then issuing them to the next stage of a pipeline, in order to achieve CPI = 0.5 without a guarantee of CPI uniformity.

In Figure 7, the behavior of the original PicoBlaze versus a standard two-stage pipeline versus our proposed method is shown. It assumes that instructions in program

memory are numbered from 1 upwards: 1, 2, 3, and so on. FDx stands for fetch/decode of instruction number *x*, and Ewx stands for execute and write back of instruction number *x*. Tax means instruction located at target address x, and EWTAx means execution and write back of Tax.

Figure 7 demonstrates the branch delay elimination. It shows how a standard pipeline stalls when a branch instruction is taken. The already fetched and decoded instruction, number 3 (FD3), must be discarded, and its execution and write back stages (EW3) cannot be performed if instruction 2 is a jump to instruction 9 and the branch is taken. This results in an invalidated pipeline and imposes a one clock cycle penalty. In contrast, our proposed method never generates any stalls. Notice that although the clock period of the proposed processor (DAP-Zipi8) is longer than the PicoBlaze, it can execute instructions 1, 2, and 9 in shorter time (notice the end of the EW9 cycle).

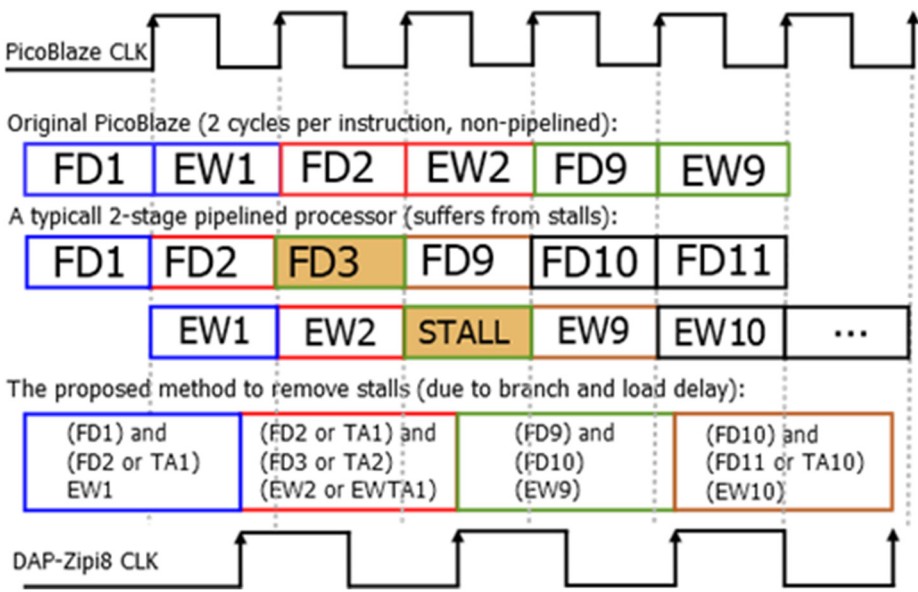

**Figure 7.** Original PicoBlaze versus a two-stage pipeline versus the proposed method (assuming instruction 2 is a conditional branch to arbitrary location 9 and it is taken).

Figure 8 demonstrates the load delay elimination. In normal flow, one instruction is fetched per clock, and if there is a data dependency between the current fetched instruction and the previous instruction, then a stall must occur. The proposed dual-fetch mechanism leads to complete elimination of delays related to data dependency among two consecutive instructions. For example, if inst_1 depends on the inst_0 result saved in register A, then a forward path will send on the result of inst_0 to inst_1. The detection is simply done by comparing the source and target registers of the instructions.

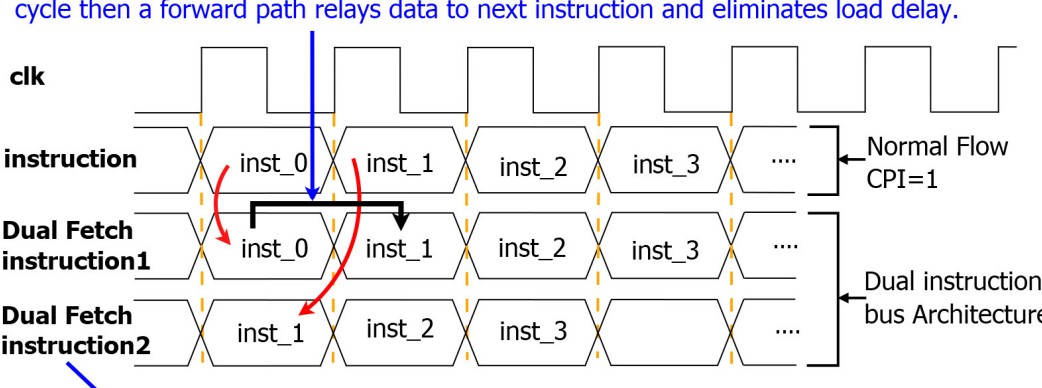

**Figure 8.** Description of the dual-fetch mechanism showing load delay elimination using two instruction buses.

For PicoBlaze, two clock cycles per instruction are necessary. In the first clock cycle, an instruction from the location address pointed to by the PC register is fetched. A second cycle is required to decode and execute the fetched instruction. The write back happens in the same clock cycle while the next instruction fetch occurs. This second cycle is mandatory for three kinds of instructions: conditional jump, 'return', and 'call@(sY, sY)' instructions. The reason is that the next PC value depends on other signals, such as zero/carry flags, the stack, or register content. Therefore, the design by Xilinx opts for two clocks per instruction; one clock cycle to "fetch and write back" and another to "decode, execute, and calculate next PC value". This yields uniform ISA with CPI = 2 for all instructions. The search for how to reduce CPI while keeping ISA *uniformity* intact motivated the work presented in the next section.

### 7. Zipi8 (CPI = 2) to DAP-Zipi8 (CPI = 1)

The first step is to set program memory BRAM to dual-port mode with the following settings:

- Memory Type = "True Dual-Port RAM";
- Primitives output Register = "Unchecked".

Apart from *address* and *instruction* buses, two more buses named *address2* and *instruction2* are added to fetch an extra instruction on every rising edge of the clock. The original design updates the PC signal every two cycles based on control signal *t_state(1)* and is toggled every cycle.

By removing the *t_state(1)* signal, the PC value is forced to be updated every clock cycle. The next step is to remove all D flip-flops (FDs) which take part in the construction of the two-stage pipeline. All modifications applied to all 16 modules of the Zipi8 core are listed in Supplementary S4.

After applying the changes, a single-cycle processor is nearly achieved. It includes fetch, decode, and execution stages all in one cycle. However, the new design fails to calculate the correct next pc value if the processor state machine deviates from the normal flow (*pc* = *pc_value* + 1). Figure 9 elaborates this failure when normal flow is disrupted by a branch instruction. Let us assume that an instruction at memory location 0x002 is a conditional jump to an arbitrary target memory address 'x'. The processor fetches inst_0 and inst_1 from memory location 0x000 and 0x001 as normal. The PC value is then set to 0x002 and, in the next clock cycle, the jump@x instruction is fetched. As the design still needs two clock cycles to calculate the right *pc_value*, the jump target address value propagates to *pc* one clock cycle late and the instruction after the conditional jump (inst_3, which should not be reached by the processor and inst_2, is a jump and is taken)

is then fetched wrongly. This is the inherent problem of branch instructions that leads to pipeline stalls as discussed earlier.

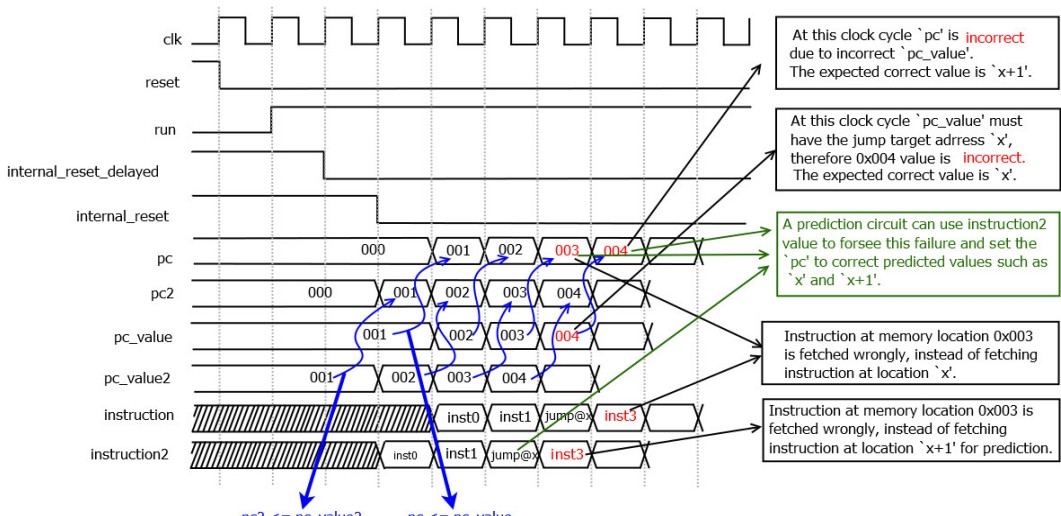

**Figure 9.** Elaboration of Zipi8 instruction fetch failure after its modifications to achieve CPI = 1 and before adding a branch prediction circuit.

### 7.1. Adding the Dual Address Bus and Branch Prediction Circuit

The main idea behind the dual address bus branch prediction (DAP) circuit is to fetch two instructions per clock cycle by using a dual-port program memory block. This allows the circuit to predict the next PC value correctly by decoding the first fetched instruction in one clock cycle and then using the decoded signals in the execution step of the next clock cycle.

The schematic provided In Figure 10 shows the Zipi8 modules which must be modified to accommodate the DAP circuit (added signals are in blue). Note that ALU, decoder, and SPM modules are not shown in this figure as these modules remain intact. The most important added signals are *instruction2* and *address2*. They are connected to the second port of external program memory BRAM. The *address2* signal (derived by *pc2*) holds the address of the second instruction and is always fetched in parallel with the current instruction (derived by *pc*). Both *pc* and *pc2* are generated by the 'Program Counter' module (module no. 7 and 16).

The second most important modification is the conversion of RAM32M primitives to dual-port instances in module no. 7, 16, and 13. This enables two locations of stack memory and registers (which are also memory) to be accessed instead of one location. This is achieved through simultaneous access to PORTA and PORTB of block RAMs on every clock cycle.

This is mandatory for prediction of the target address of instructions such as 'return', which uses the stack memory (using PORTB of the stack memory), or 'call@(sX, sY)', which uses the register banks (using PORTB of the register bank memory). sx_addrB and sy_addrB are connected to PORTB of the register memory BRAM, which makes simultaneous access to 2 out of 32 registers possible through PORTA and B.

Referring to Figure 10, the *sxB* and *syB* signals are the output of BRAM's PORTB in general-purpose register banks. The *push_stack2* and *pop_stack2* signals, alongside the *pc2* signal, are added to the PORTB of BRAM used in the 'Stack' memory module. These signals assist the prediction of correct stack pointer values and, consequently, the 'Stack' module can then set the correct 'stack_memory' value, as well as other necessary outputs.

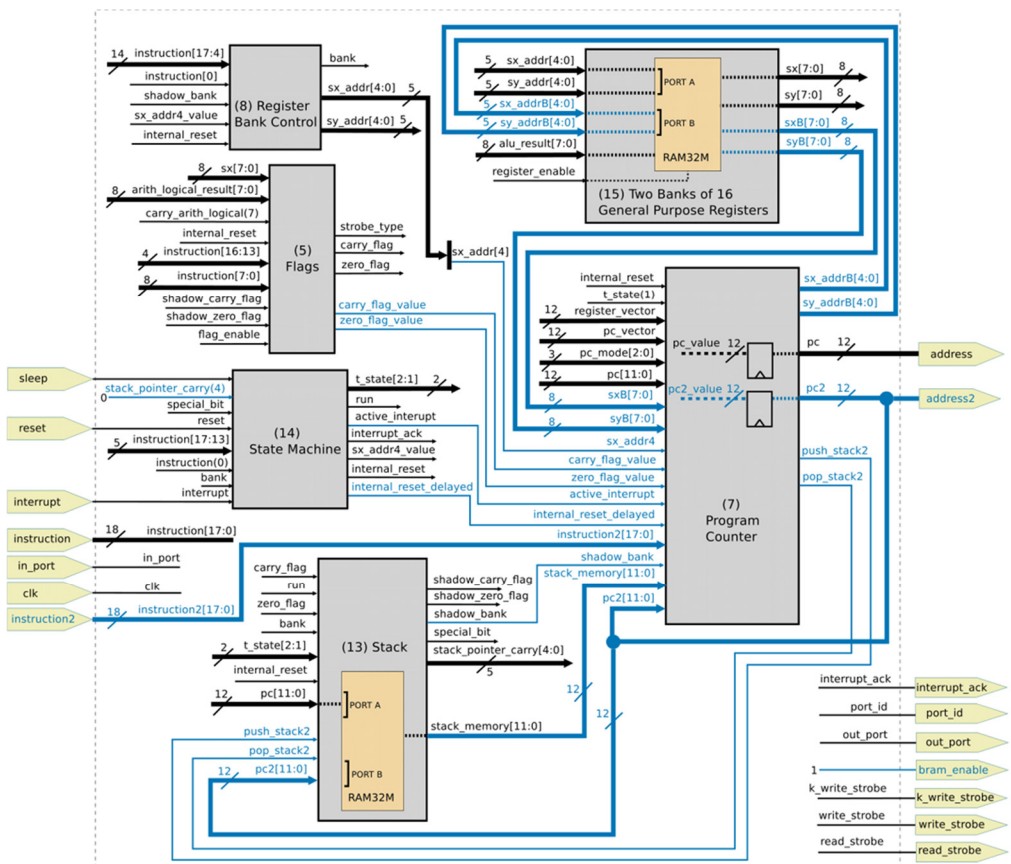

**Figure 10.** Zipi8 schematic with added prediction signals (highlighted in blue)—ALU, decoder, and SPM modules are omitted as these modules remain unchanged.

Next is the addition of the *internal_reset_delayed* signal to the 'State Machine' module. As can be seen in Figure 9, this signal goes low one clock cycle earlier than the *internal_reset* signal. This provides one extra clock cycle to the 'Program Counter' module for predicting the *pc2* value. The *carry_flag_value* and *zero_flag_value* (see Figure 10) are simply the next values of the *carry_flag* and *zero_flag* signals calculated based on the execution of the current instruction. There are few internal signals of the 'Flags' module that are required to be routed out of the module for use as input to the 'Program Counter' module for prediction. In the original design, the 'Register Bank Control' module is responsible for producing *sx_addr* and s̲y̲_̲a̲d̲d̲r̲ and depends on the *sx_addr4_value* produced by the 'State Machine' module. The purpose of reusing 'Register Bank Control' and moving it into the 'Program Counter' module is to generate the *sx_addr [4]* signal. The core of the prediction mechanism is inside the 'Program Counter' module, which will be discussed in the next section.

### 7.2. Program Counter Module Modification

In the original PicoBlaze, the 'Program Counter' module is responsible for determining the next PC value in each clock cycle. Figure 11 depicts the internal structure of the modified 'Program Counter' module. Analysis of PicoBlaze shows that the 'Program Counter' module receives the following signals as input:

1. *pc* (current state);
2. *register_vector*;
3. *pc_vector*;
4. *pc_mode* (current inputs).

The module then calculates the *pc_value* as output, which is then clocked to the PC. This constructs a simple *Mealy* state machine where the output depends on inputs and the

current state of machine. The state machine then identifies the four necessary signals that must be present to calculate the next PC register value (*pc_value* signal).

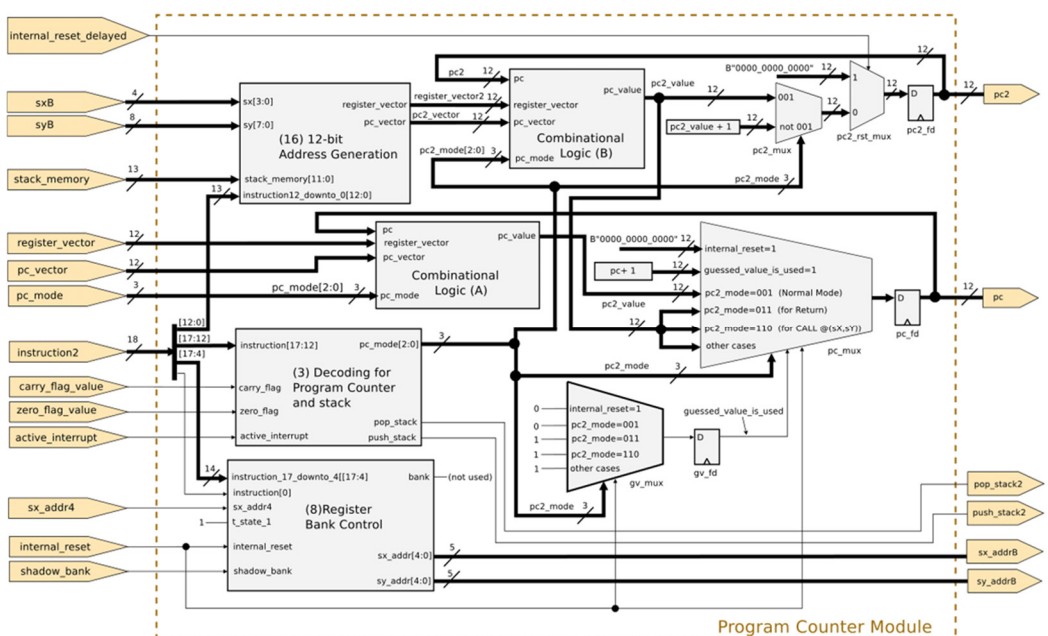

**Figure 11.** Program Counter module with the prediction circuit added (a high-resolution version of this figure is available in Supplementary S3).

At the heart of Figure 11 we have 'combinational logic (A)', which receives the *pc*, *register_vector*, *pc_vector*, and *pc_mode* values and generates the *pc_value* which is the next value of the PC register. This block is combinatorial and, in the original design, is constructed using LUT6, MUX, and XOR primitives. The exact duplication of this block is named 'combinational logic (B)' and is used to generate the *pc2_value* signal. The inputs to combinational logic (B) are derived from exact duplication of '(16) 12-bit Address generation' and '(3) Decoding for program counter and stack' modules. Instead of *instruction*, *sx*, *sy*, *carry_flag*, and *zero_flag* signals, the *instruction2*, *sxB*, *syB*, *carry_flag_value*, and *zero_flag_value* signals drive their inputs. This produces the *pc2_value* signal which is the potential guessed value for the next PC register value.

Three modes are defined based on the two fetched instructions A and B, and the details of how the final value of the PC register is calculated and set are then discussed. The modes are:

1. 'Normal' mode: instructions A and B do not modify the PC register and both are not JUML, CALL, or RETURN instructions.
2. 'Guessed value is used' mode: instruction B modifies the PC register but instruction A does not.
3. 'Illegal' mode: instructions A and B both modify the PC register.

In the original design, the *pc_value* signal (the next value of the PC register) is directly connected to the 'pc_fd' flip-flop. The design is modified by adding the 'pc_mux' multiplexer before the 'pc_fd' flip-flop, which selects the correct predicted *pc_value* based on three signals ordered from high to low priority:

1. *internal_reset;*
2. *guessed_value_used;*
3. *pc2_mode.*

If the *internal_reset* signal is high, regardless of other multiplexor selectors, the *pc* will be set to zero (processor reset). If the *internal_reset* signal is low, then the processor is in running mode and the *guessed_value_used* signal will be checked. When

*guessed_value_used* is high, it means the processor is in 'Guessed value is used' mode, which indicates that current instruction A has modified the PC register and, consequently, the guessed value has been used already; therefore, the next valid instruction will be in the '*pc* + 1' memory location. Note that addition of this multiplexor increases the critical path of the processor.

It should be noted that it is illegal to have two consecutive instructions which both modify the PC register. Therefore, if the current instruction has modified the PC, the assumption is that the next one will not; therefore, incrementing the PC by one is always the correct way to advance the processor state machine. When *guessed_value_used* is low, it means the processor is in 'Normal' mode, which indicates that the current instruction does not modify the PC register.

The next step is to investigate the next instruction that has already been fetched and decoded. The binary value '0b001' for the *pc2_mode* signal indicates that the next instruction will not modify the PC register and, therefore, that *pc_value* is the next value of *pc*. The binary value '0b011' for the *pc2_mode* signal indicates that the next instruction is a RETURN instruction; therefore, *pc_value* must be discarded and, instead, the return address fetched from stack in advance (*pc2_value*) must be used as the next *pc* value. The binary value '0b110' for *pc2_mode* indicates that the next instruction is a 'CALL@(sX, SY)' instruction and that the next value of *pc* must be a concatenation of the content of [xS,xY] registers, both of which are fetched from the register bank in advance and placed on the *pc2_value* signal.

In Figure 11, the *pc2_mux* and *pc2_rst_mux* signals select the next value for the *pc2* signal. If the *internal_reset_delayed* signal is high (processor reset), then *pc2* will be set to zero; otherwise, the next value of *pc2* will be either *pc2_value* (Normal mode) or *pc2_value* + 1 (not Normal mode).

The last module that needs to be discussed here is the 'Register Bank Control' module, which outputs the *sx_addr [4]* signal. As shown in Figure 11, input signal *sx_addr [4]*, alongside the *shadow_bank* and i*nstruction2* signals, sets the output signals *sx_addrB* and *sy_addrB*. These two output signals hold values destined for PORTB of the register bank's memory.

To summarize the modification technique presented above, all memory blocks are converted to dual-port mode to fetch two instructions in parallel. The minimum logic in the original decoder is then replicated ('combinational logic (A) and (B)') to produce signals for prediction circuitry that result in removal of branch and load delays.

### 7.3. Stack Module Modification

In the original PicoBlaze design, the 'Stack' module is responsible for producing *zero_flag*, *carry_flag*, *bank*, and *special_bit* signals alongside of the *stack_memory* signal, as detailed in Supplementary S2. They depend on *push_stack* and *pop_stack* input signals set by decoding circuitry. The current value of the internal signal *stack_pointer* drives the ADDRA port of BRAMs that is used as stack memory. The memory content the *stack_pointer* points to holds the return value address.

For example, if the processor executes a RETURN instruction, then a *pop_stack* signal will be asserted which prompts the 'Stack' module to decrement *stack_pointer* by one. This will put the memory content of "*stack_pointer* − 1" on the stack memory output data bus, which in turn recovers the *flags*, *bank*, and *pc* register values. When a CALL instruction gets executed, the *push_stack* signal is asserted, which prompts *stack_pointer* to be incremented by one ("*stack_pointer* + 1"). Next, the *WE* signal will be set to a high value so the current flags and *pc* value can be saved into stack memory.

The modification of the original design starts by enabling the dual-port option for BRAMs used as stack memory. The *push_stack* and *pop_stack* inputs must be removed as they are calculated one clock cycle late (PicoBlaze uses two clock cycles, and these two signals are used in the second clock cycle). These two input signals are replaced by *push_stack2* and *pop_stack2* signals, which are generated by prediction circuitry in ad-

vance. They detect whether the current instruction is a RETURN or a CALL, which prompts a pop from stack memory or a push into stack memory, respectively.

The next step is the removal of all LUT, MUX, XOR, and FD primitives and redesign of the 'Stack' module to accommodate prediction circuitry, which is shown in Figure 12. The *stack_pointer* signal is connected to the ADDRA port, and a series of multiplexors decide whether the pointer must be incremented or decremented based on the values of *push_stack2* and *pop_stack2*. The *stack_pointer* is connected to the ADDRB port and always points to the location of "*stack_pointer* − 1".

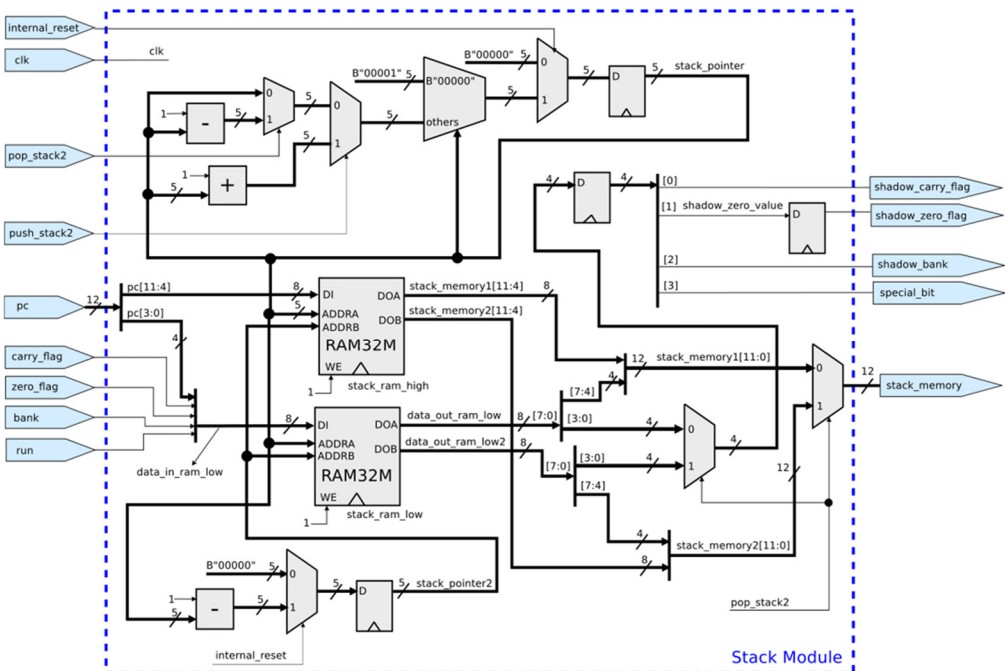

**Figure 12.** Stack module with prediction circuit added (a high-resolution version of this figure is available in Supplementary S3).

This makes the content of memory locations at *stack_pointer* and "*stack_pointer* − 1" available at any given clock cycle through *stack_memory1* (memory content on ADDRA) and *stack_memory2* (memory content on ADDRB) signals. The two multiplexors, with *pop_stack2* as their selector, decide the final value of the *flag*, *bank*, and *stack_memory* signals. Note that the *WE* pins of both BRAMs are permanently pulled up (connected to Vcc), which forces a continuous write on every clock cycle at the memory location pointed to by the ADDRA value.

With addition of the circuit mentioned above, the processor constantly writes the status of flags and the current PC register value into stack memory at every clock cycle (constant push). At the same time, it constantly reads two locations from stack memory pointed to by the stack pointer and the stack pointer subtracted by one. The prediction circuit drives the processor to pop the stack (decrement the stack pointer by one and use the output of PORTB to recover the PC register value and flags through the *pop_stack2* signal) or continue normal operation (the stack pointer will be intact and the output of PORTA will be used). In case of a push, the processor just needs to increment the pointer by one (triggered by *push_stack2*), as writing to stack memory is already performed on every clock cycle regardless of whether the *push_stack2* signal is asserted.

## 8. Resource and Power Utilization

Table 3 compares the resource utilization of our proposed DAP-Zipi8 with CPI = 1 with that of Zipi8 with CPI = 2 and the original PicoBlaze. Referring to Table 3, the max-

imum clock frequency obtained with the Xilinx Zynq UltraScale+ MPSoC ZCU104 Evaluation Kit was 369.041 MHz, which is attributed to the original Xilinx PicoBlaze.

The conversion of a firm-core PicoBlaze to a soft-core Zipi8 is essential if we want to modify the design. The converted soft core, named Zipi8, can achieve a maximum frequency of 357.509 MHz (=2.86% decrease) and an increase in LUT count from 122 to 157 (28.69% increase). This is the cost (increase in logic area) that must be paid to make the register transfer level (RTL) HDL source code of PicoBlaze available for modification.

**Table 3.** PicoBlaze vs. Zipi8 vs. DAP-Zipi8 resource utilization and maximum clock frequency on a Xilinx ZCU104 development board.

| Core | CPI | Max Freq. (MHz) | LUTs | Regs. | Carry8 | F7 Mux | F8 Mux |
|---|---|---|---|---|---|---|---|
| PicoBlaze (KCPSM6) | 2 | 369.041 | 122 | 74 | 7 | 16 | 8 |
| Zipi8 | 1 | 357.509 | 157 | 74 | 0 | 16 | 8 |
| DAP-Zipi8 | 1 | 224.022 | 305 | 49 | 2 | 16 | 8 |

The dual-fetch technique, explained previously in conjunction with dual-port memory, and the addition of dual-address bus prediction circuitry yields a new processor that is named DAP-Zipi8, which has a LUT count of 305 (94.27% increase compared to Zipi8) and a maximum frequency of 224.022 MHz on the Xilinx ZCU104 development board. Note that the removal of flip-flops between decoder and execution stages results in a reduction in total register count from 74 to 49 (see Table 3).

Although the DAP-Zipi8 critical path has increased (which results in lower achievable maximum clock frequency), CPI is reduced from two to one (50% decrease). Considering processor performance in terms of million instructions per second (MIPS), the calculation of MIPS for Zipi8 (CPI = 2) is shown in Equation (3).

$$\frac{Max\ Freq.}{CPI} = \frac{357.509 MHz}{2} = 178.75\ MIPS \tag{3}$$

For DAP-Zipi8, the calculation is shown in Equation (4).

$$\frac{Max\ Freq.}{CPI} = \frac{224 MHz}{1} = 224\ MIPS \tag{4}$$

MIPS is not an accurate metric to compare the performance of processors with different ISAs. For example, comparison of Intel versus ARM versus PicoBlaze using the MIPS metric is incorrect. In those cases, benchmarks from Dhrystone, the Embedded Microprocessor Benchmark Consortium (EEMBC), or the Standard Performance Evaluation Corporation (SPEC) should be used, as instructions in different ISAs might perform different amounts of work. For example, one ISA might have an instruction that performs a simple addition operation, while another ISA might have special digital signal processing (DSP) instructions that perform addition and multiplication as one combined instruction. In this paper, the modified PicoBlaze (DAP-Zipi8) and PicoBlaze itself both have the same ISA and are compared against each other. This justifies employment of the MIPS metric as a meaningful performance indicator [17].

As shown in Figure 13, a 25.31% performance improvement for the DAP-Zipi8 processor (increase from 178.75 to 224 MIPS) is achieved without considering the impact of mandatory NOP instructions (inserted to avoid invalid examples of the design).

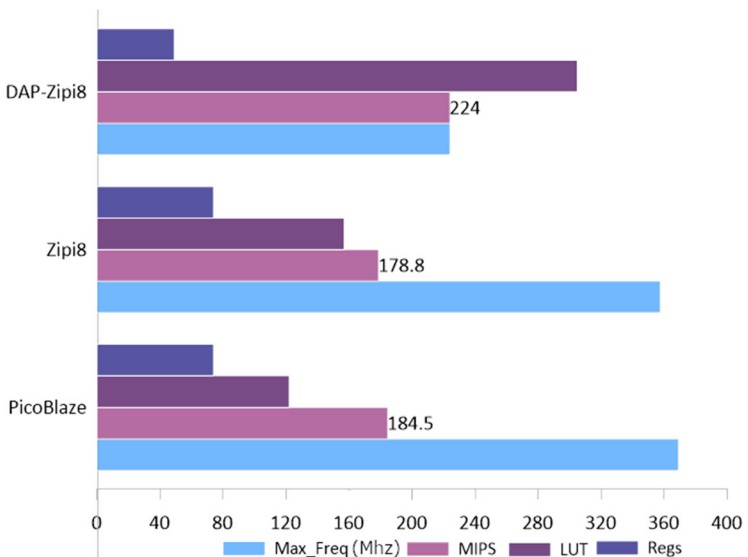

**Figure 13.** PicoBlaze vs. Zipi8 vs. DAP-Zipi8 performance and resource utilization comparison.

Figure 14 shows four programs that were executed on both PicoBlaze and DAP-Zipi8 cores and their measured execution times (left side *y*-axis). On the *x*-axis, the 'state_machine' label refers to a complex finite state machine (FSM) that is used to control 1024 PicoBlaze cores on a single FPGA chip (XCZU7EV-2FFVC1156). The FSM is written by the authors and follows the work presented in [75]. It is an RTES application that uses a myriad of cores for the adaptive routing and serving of swarms of incoming network traffic (miniature web servers running on PicoBlaze instances). Utilizing a PicoBlaze core as an FSM controller is common and the detail of this implementation, though out of the scope of this article, will be published in another paper. Note that this state machine is the main application of our proposed architecture, and it is not executable on indeterministic processors (processors with a pipeline or caches) as it expects programs to be executed in an exact predetermined number of clock cycles. It is noteworthy that the worst-case response time to external interrupts in this state machine is just three clock cycles using DAP-Zipi8 versus five cycles in the original PicoBlaze.

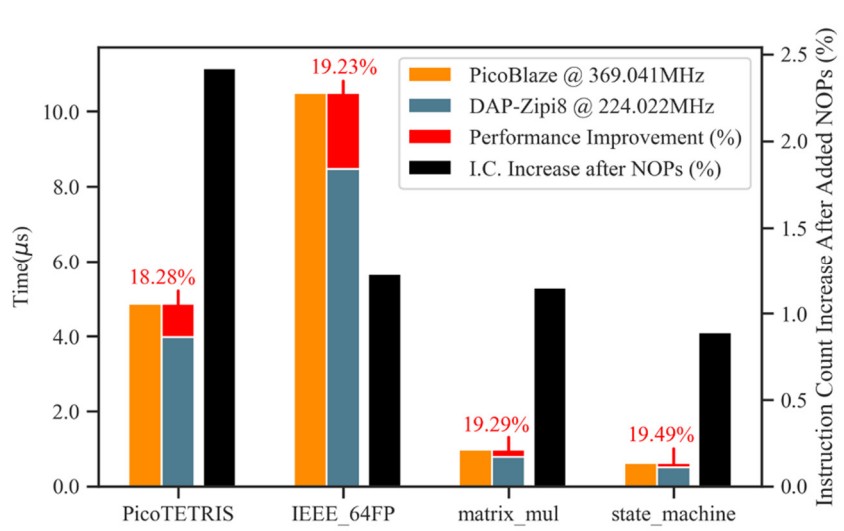

**Figure 14.** PicoBlaze vs. Zipi8 vs. DAP-Zipi8 performance and resource utilization comparison.

To avoid the invalid case ('Illegal' mode as defined in Section 7.2) where two consecutive conditional jump instructions are present, a NOP instruction is inserted in between of jump instructions. A program in the final pass of compilation scans program opcodes and inserts a NOP instruction whenever two consecutive branch instructions are discovered.

In Figure 14, the negative impact of added NOP instructions on performance for all algorithms is shown. The algorithms used in the benchmark are listed below:

1. PicoTETRIS [76]: a Tetris game written in PicoBlaze assembly language.
2. An IEEE-754 64-bit floating point arithmetic library [77].
3. A matrix multiplication algorithm written in PicoBlaze assembly language.
4. A state machine controller that manages multiple cores by implementing a complex FSM used to controlled 1024 PicoBlaze cores on an FPGA in real time.

The added NOPs increase the total instruction count (right-side *y*-axis in Figure 14 which is labelled "Instruction Count Increase after Added NOPs") per algorithm. The percentage increase is shown with the black bar along the right-side *y*-axis. Although the number of NOP instructions increased by around 1% (2.42% in the case of PicoTETRIS), their impact was quite significant. It reduced the performance gain from the initially obtained 25.31% down to 18.28~19.49%, as reported in Figure 14.

Power consumption was measured using the Xilinx Vivado v2021.1 'Power Report' facility for all three cores at both 100 MHz clock frequency and maximum achievable clock frequency. The total FPGA on-chip power consumption for all cores was 722~737 mW, which is divided into static and dynamic power. Static power was fixed at 615 mW and was FPGA device dependent. Total dynamic power consumption was 107~122 mW. A large portion of dynamic power was used in memory block RAM, clock generation circuitry, and other support modules. The portion of dynamic power that was used by cores is reported in Table 4. It shows a 42.86% power increase (against 25.31% performance gain) in DAP-Zipi8 when running the cores at their maximum frequency.

**Table 4.** PicoBlaze vs. Zipi8 vs. DAP-Zipi8 power utilization on a Xilinx ZCU104 development board.

| Core | Power @ 100 MHz | Power @ Maximum Achievable Frequency |
|---|---|---|
| PicoBlaze (KCPSM6) | 3 mW | 12 mW |
| Zipi8 | 4 mW | 14 mW |
| DAP-Zipi8 | 8 mW | 20 mW |

## 9. Verification

Three verification methods have been employed to ensure the correctness of DAP-Zipi8 and its compatibility with PicoBlaze.

### 9.1. Isolated Instruction Execution

To verify the DAP-Zipi8 core, execution results of all instructions on the machine state (registers, flags, scratch pad memory content) were examined. Each instruction was used in a tiny test program, and its execution outcome was then compared and verified by examining the simulation waveform manually. Note that the *comparison* method mentioned in Section 5.4 cannot be used here as DAP-Zipi8 is cycle-incompatible with PicoBlaze. Correct results were observed by thorough examination of waveforms.

### 9.2. Math Library Execution

To verify conditional jumps, calls, returns, and stack mechanisms, both cores were set to execute a sequence of complex math methods. The results of those methods were obtained and then compared by running the code on both cores. The IEEE-754 64-bit floating point arithmetic library [77], which has relatively high complexity, was used to expose potential faults. The library has numerous bitwise operations that boost the

chance of discovering processor bugs, even if a single bit is miscalculated as it tries to compute 64-bit floating point numbers on an 8-bit processor.

The library extensively uses 8-bit registers and scratch pad memory to perform 64-bit normal/subnormal floating point (FP) operations. The math methods use carry and zero flags for almost every routine and are extremely sensitive to any mistakes while performing bit slicing, shifting, concatenation, and other bitwise operations. Figure 15 shows the hardware setup used to verify DAP-Zipi8.

The Xilinx ZCU104 development board has a Zynq Ultrascale+ chip which hosts a hardened ARM Cortex-A53 processor. The ARM core has an internal floating-point unit (FPU) and can perform IEEE-754 64-bit operations natively. The core performs floating-point operations and the operands used in calculations, as well as the results, are saved in a dual-port shared BRAM. DAP-Zipi8 was also connected to this dual-port BRAM and its reset pin was controlled by the ARM processor. After the reset signal was asserted by the ARM core, the DAP-Zipi8 was reset and then read the requested FP operation stored in the shared BRAM. Next, the appropriate library routine that matched the requested operation was called. The result produced by DAP-Zipi8 was then saved back into the shared BRAM and an interrupt signal was sent back to the ARM core to signal the end of the operation. The ARM core then fetched the result produced by DAP-Zipi8 and simply compared it with its own result and printed an error message if a mismatch in results occurred.

Initial runs of this verification method revealed several bugs. After addressing the bugs, no inconsistencies were found between the results for both cores. Knowing that hardened ARM floating point units are bug-free, DAP-Zipi9 is also concluded to contain no mistakes.

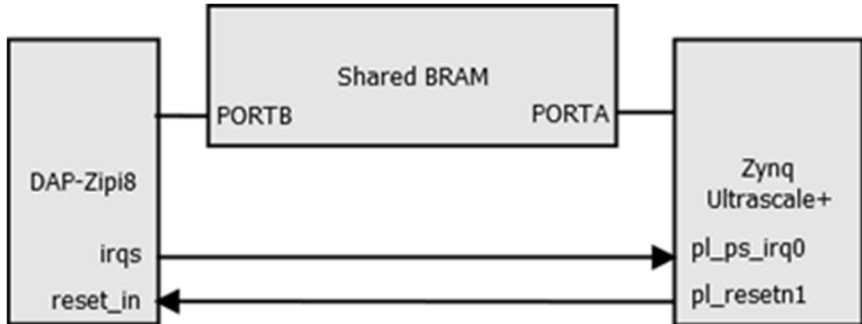

**Figure 15.** Hardware setup for DAP-Zipi8 verification using a math library on a Xilinx Zynq Ultrascale+ ZCU104 development board.

*9.3. Random Execution from an Instructon Pool*

A C++ program was developed by the authors to generate a series of PicoBlaze instructions randomly. Figure 16 shows the C++ classes used in the program. The randomly generated instructions from an instruction pool were passed to both cores (PicoBlaze and DAP-Zipi8) for execution. After the completion of execution simulation (with random instructions loaded into BRAM program memory), the final states of both cores were compared. A bug was reported if there was a discrepancy in register content or flag status. Results obtained in this step reinforce the correctness of the design.

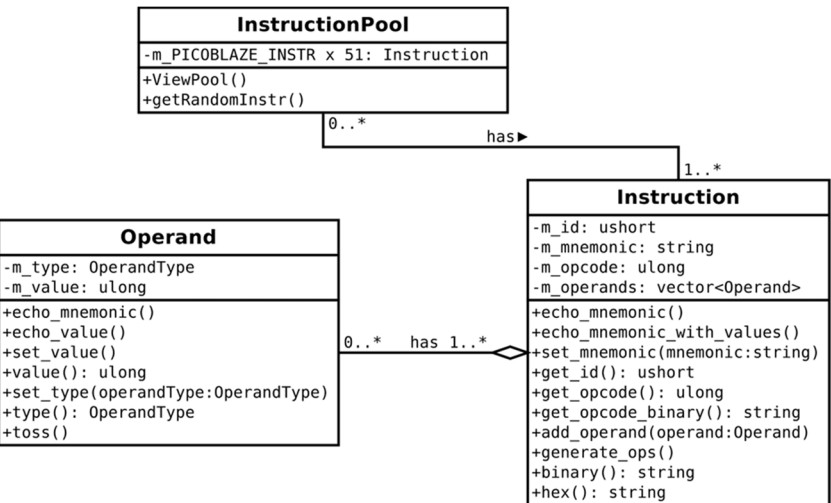

**Figure 16.** C++ classes for randomly generating PicoBlaze instructions.

## 10. Conclusions

In this paper, DAP-Zipi8, which is an 8-bit PicoBlaze compatible soft core, is proposed. It is equipped with a branch prediction that uses two address buses to calculate branch targets in order to eliminate load and branch delays. By adapting the new design, DAP-Zipi8 exhibits a performance boost from 178.75 MIPS to 224 MIPS (25.31%) compared to the original PicoBlaze. The case of two consecutive conditional branches is the only invalid situation, which can be easily avoided by inserting NOP instructions between branch instructions. After measuring the negative impact of extra added NOP instructions, an 18.28–19.49% increase in overall core performance was achieved. The improved performance is a trade off with increased LUT count (157 to 305 LUTs, which is a 94.27% increase in logic area). The enhanced DAP-Zipi8, with its deterministic ISA, emerges as a good candidate for hard RTESs. The core is used as a real-time state machine controller for a homogenous multi-core architecture (1024 PicoBlaze cores).

**Supplementary Materials:** The code associated to the work presented in this paper is VHDL source code for the DAP-Zipi8, an improved implementation of the Xilinx PicoBlaze, and can be found in Supplementary S1. The Supplementary Materials are available on the public GitHub repository: https://github.com/ehsan-ali-th/DAPZipi8Appendices. Supplementary S1: VHDL source code of DAP-Zipi8 Xilinx Vivado 2020.1 Project: https://github.com/ehsan-ali-th/DAPZipi8Appendices/tree/main/Appendix_A (accessed on 23 October 2022). Supplementary S2: Complete high-resolution schematic of Zipi8 in E PS format: https://github.com/ehsan-ali-th/DAPZipi8Appendices/tree/main/Appendix_B (accessed on 23 October 2022). Supplementary S3: High-resolution version of Figure 4, 11, and 12. https://github.com/ehsan-ali-th/DAPZipi8Appendices/tree/main/Appendix_C (accessed on 23 October 2022). Supplementary S4: Zipi8 modules modifications for improving CPI to 1. https://github.com/ehsan-ali-th/DAPZipi8Appendices/tree/main/Appendix_D (accessed on 23 October 2022).

**Author Contributions:** All work presented in this paper was performed by E.A. under the supervision of W.P. All authors have read and agreed to the published version of the manuscript.

**Funding:** This research is supported financially by 'The Chulalongkorn Academic Advancement into Its 2nd Century Project'. A joint research fund, composed of 'The 100th Anniversary Chulalongkorn University Fund for Doctoral Scholarship' and 'The 90th Anniversary of Chulalongkorn University, Rachadapisek Sompote Fund', was received.

**Data Availability Statement:** All data related to this article can be found at https://github.com/ehsan-ali-th/DAPZipi8Appendices.

**Conflicts of Interest:** The authors declare no conflict of interest. The funders had no role in the design of the study; in the collection, analyses, or interpretation of data; in the writing of the manuscript; or in the decision to publish the results.

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
