# Peer review of "A Deterministic Branch Prediction Technique for a Real-Time Embedded Processor Based on PicoBlaze Architecture"

_electronics, doi:10.3390/electronics11213438_

Round 1

Reviewer 1 Report

Please,find the review in the attached file

Author Response

The response is included in attached MS Word file.

Reviewer 2 Report

The author presented a novel method for branch prediction for central processing units in the article. The proposed technique helps to avoid the possible problems caused by the conditional jump instructions. Overall, the paper is interesting, and the results look good. However, there are a few comments as follows:

1. In Figure 11, the author presents the diagram for the modified Zipi8. However, there is no ALU component in this. It is better to have a full diagram as Figure 4 with highlighting the differences.

2. Figure 13 shows the Stack module architecture. However, the caption for this is the same as Figure 12. Please, clarify this.

3. CPI value in Equation (4) is wrong, it should be “1” in the case of using DAP-Zipi8 architecture.

4. Quality of Figure 1 should be improved.

5. Caption and content of the same table (Tables 1 and 5) need to be in the same page.

Author Response

(The authors gave the same response as above.)

Reviewer 3 Report

Major remarks

    + Figure 5 is difficult to understand, I suggest to highlight only the relevant parts of the schematic and to remove the circuitry that is not directly referenced in the text, to help readability;
    + To me it is not completely clear why the authors chose the Xilinx Picoblaze to investigate their contributions, instead of an opensource core. Maybe I can suggest to detail better this aspect in the introductory or background sections;

    + The experimental section that shows the behaviour of the proposed system in a real context is limited. What I suggest is to consider a much broader experimental campaign, in order to better appreciate the benefits obtained. Currently only four benchmarks are considered. I also recommend the use more representative benchmarks in the context of embedded/real-time domain. For example embenches (https://www.embench.org/) could be more adequate;

    + The final goal of the proposed work is to modify the design of a core in order to make its behavior predictable. However this aspect is not addressed and validated in the experimental section. To this end, a type of experiment that I recommend to take into consideration is to quantify the variability on the execution time of a benchmark, considering the standard Picoblaze, compared to the modified version;

    + Section 7 which introduces the design changes in order to obtain a predictable CPI = 1 is perhaps too long and unbalanced compared to the rest of the paper. For this reason it is rather complex to follow that section in all its parts. I suggest to slightly reduce this part, eventually moving part of this in the appendix and use the space obtained for further experiments.

Minor remarks

    + "no benchmarking program is required because they both cores.." --> "no benchmarking program is required because both cores..";

    + The instruction "ADD 1 s0", I think that should be "ADD s1, s0";

    + In the "definitions" section at page 2, the authors say that for the implementation of a processor-based design, in addition to FPGA and ASIC devices, it is possible to use MCUs. However, it is not entirely clear what is meant in this case, since MCUs are fixed hardware devices, that can not be re-configured like FPGAs.

Author Response

(The authors gave the same response as above.)

Round 2

Reviewer 1 Report

All my comments are taken into account. The paper is corrected in a proper way. I think that now the artcile can be accepted in the current form.